# Synthesis of Co_3_O_4_ Nanoparticles-Decorated Bi_12_O_17_Cl_2_ Hierarchical Microspheres for Enhanced Photocatalytic Degradation of RhB and BPA

**DOI:** 10.3390/ijms232315028

**Published:** 2022-11-30

**Authors:** Syed Taj Ud Din, Wan-Feng Xie, Woochul Yang

**Affiliations:** 1Department of Physics, Dongguk University, Seoul 04620, Republic of Korea; 2School of Electronics and Information, University-Industry Joint Center for Ocean Observation and Broadband Communication, Qingdao University, Qingdao 266071, China

**Keywords:** Co_3_O_4_/Bi_12_O_17_Cl_2_, heterojunction, photocatalysis, Rhodamine-B, Bisphenol-A

## Abstract

Three-dimensional (3D) hierarchical microspheres of Bi_12_O_17_Cl_2_ (BOC) were prepared via a facile solvothermal method using a binary solvent for the photocatalytic degradation of Rhodamine-B (RhB) and Bisphenol-A (BPA). Co_3_O_4_ nanoparticles (NPs)-decorated BOC (Co_3_O_4_/BOC) heterostructures were synthesized to further enhance their photocatalytic performance. The microstructural, morphological, and compositional characterization showed that the BOC microspheres are composed of thin (~20 nm thick) nanosheets with a 3D hierarchical morphology and a high surface area. Compared to the pure BOC photocatalyst, the 20-Co_3_O_4_/BOC heterostructure showed enhanced degradation efficiency of RhB (97.4%) and BPA (88.4%). The radical trapping experiments confirmed that superoxide (^•^O_2_^−^) radicals played a primary role in the photocatalytic degradation of RhB and BPA. The enhanced photocatalytic performances of the hierarchical Co_3_O_4_/BOC heterostructure are attributable to the synergetic effects of the highly specific surface area, the extension of light absorption to the more visible light region, and the suppression of photoexcited electron-hole recombination. Our developed nanocomposites are beneficial for the construction of other bismuth-based compounds and their heterostructure for use in high-performance photocatalytic applications.

## 1. Introduction

Rapid industrialization and population growth has led to a tremendous increase in environmental pollutions. These pollutants mostly consist of hazardous Azo dyes and phenolic compounds. Rhodamine-B (RhB) cationic Azo dye is an anthraquinone derivative. It is highly stable and non-biodegradable in nature and is classified as a carcinogenic and neurotoxic substance [1]. Aside from dyes, the other frequently used compound is the colorless Bisphenol-A. It is a diphenylmethane derivative and a raw material that is widely used in the fabrication of numerous polymeric materials [2]. Long-term exposure of BPA causes endocrine, neurological, and reproductive developmental disorders [3]. Therefore, it is crucial to eradicate RhB and BPA before waste is discharged into water reservoirs and landfills. Pollution-free environmental remediation technologies to degrade these organic pollutants have attracted substantial attention [4]. Among them, visible-light-driven photocatalytic technology has emerged as the most promising approach for wastewater cleaning and pollutant removal [5].

Recently, bismuth-based nanomaterials, such as BiPO_4_ [6], Bi_2_O_2_CO_3_ [7,8,9], Bi_4_Ti_3_O_12_ [10], Bi_2_MoO_6_ [11], Bi_2_WO_6_ [12], Bi_2_O_3_ [13], and BiOX (X = Cl, Br, I) [14,15], have attracted substantial attention for their use in photocatalytic applications, because O 2p and Bi 6s valence band hybridization not only narrows the bandgap but also enhances the mobility of photo-generated holes in the valence band. Similarly, a bismuth and oxygen-enriched bismuth-oxyhalide (Bi_12_O_17_Cl_2_ (BOC)) is a typical tetragonal phase compound composed of a layered structure with an alternate stacking of [Bi_2_O_2_]^2+^ sheets interleaved with [Cl]^−^ groups, and it represents an important class of bismuth-based photocatalysts. The photocatalytic properties of nanobelts-like BOC were first reported by Xiao et al. in 2013 [16]. Since then, there have been many research efforts focused on the preparation of BOC with different morphologies, including nanobelts, nanosheets, and flower-like morphologies. For instance, Wang and colleagues [17] prepared BOC nanobelts through a solvothermal treatment using Bi(NO_3_)_3_.5H_2_O, NH_4_Cl, and NaOH as precursors in a solvent consisting of ethylene glycol (EG) and water for the photocatalytic degradation of BPA. Liu et al. [18] presented two-dimensional (2D)-BOC nanosheets oriented along the [002] direction which showed enhanced photocatalytic RhB degradation. Similarly, Fang et al. [19] prepared 3D BOC hierarchical nanostructures using a coprecipitation method followed by calcination, and these nanostructures demonstrated high photocatalytic efficiency for RhB degradation. Among the morphologies detailed above, the flower-shaped BOC has excellent characteristics, including high surface area, good adsorption capability, and maximum light absorption. Therefore, it is essential to develop a facile method to fabricate a 3D flower-like BOC. Despite these advantages, the 3D BOC still need to resolve the issues of a rapid electron-hole recombination rate and inappropriate redox potentials. Therefore, various research strategies have been developed to overcome these issues, including the fabrication of heterojunctions [20], element doping [21], noble metal deposition [22], and graphene decoration [23]. Among these methods, heterojunction preparation is the most effective approach because of the fast transfer rate of photo-generated electron and holes (e^−^-h^+^), which facilitates the separation of the photo-generated e^−^-h^+^ pairs in the photocatalysts, which play quite an important role in enhancing the photocatalytic activity of photocatalysts. For instance, He et al. [24] obtained a BOC/𝛽-Bi_2_O_3_ composite with flower-like micro/nano architectures that demonstrated good photocatalytic activity for the degradation of 4-tert-butyphenol under visible light. In another study, Huang et al. [25] prepared BiOI@BOC heterojunction photocatalysts with high exposure of the active BiOI (001) facet, which exhibited excellent photocatalytic performance for RhB and BPA degradation. Moreover, BOC heterostructures with non-bismuth-based compounds, such as CoAl-LDH/BOC [26] and Ag_2_O/BOC *p-n* junction catalysts [27], have also been reported; both exhibit significant photo-degradation efficiency under visible light irradiation.

Co_3_O_4_ is a traditional *p*-type semiconductor (band gap, E_g_ = 1.2~2.6 eV) with interesting electronic, magnetic, sensing, and catalytic properties [28]. In particular, Co_3_O_4_-based heterojunctions have yielded high photocatalytic activity; for example, the 0D/2D Co_3_O_4_/TiO_2_ heterojunction photocatalyst has exhibited enhanced photocatalytic activity under visible light irradiations [29]. In another study, Dai et al. [30] synthesized a Co_3_O_4_/BOC photocatalyst that showed effective visible-light-driven RhB photodegradation due to the more positive value of the valence band potential of Co_3_O_4_ relative to BOC. However, BOC is an n-type semiconductor, as indicated by its positive slope in the Mott-Schottky plot [31], and the more positive valence band (VB) potential than that of the Co_3_O_4_ counterpart. Therefore, the combination of Co_3_O_4_ with BOC is favorable for the formation of the *p-n* heterojunction. As a result, the photo-generated holes on the VB of BOC could be easily transferred to the VB of Co_3_O_4_ under light illumination, thus resulting in practical separation of photo-generated e^−^-h^+^ pairs of BOC and Co_3_O_4_, which would be beneficial for a photocatalyst in terms of photo-degradation efficiency. However, to our knowledge, there is a lack of research into using a Co_3_O_4_/BOC hierarchical microsphere photocatalyst for the degradation of RhB and BPA.

Herein, the synthesis of a BOC hierarchical microsphere and its decoration with Co_3_O_4_ nanoparticles (NPs) via a solvothermal method have been reported. The heterojunction formation of the Co_3_O_4_ NPs-decorated BOC (Co_3_O_4_/BOC) was evaluated through structural, morphological, spectroscopic, and electrochemical investigations. The photocatalytic degradation efficiency of the hierarchical microsphere Co_3_O_4_/BOC heterojunction was evaluated against RhB and BPA aqueous pollutants. The results showed that the 20-Co_3_O_4_/BOC heterostructure had an outstanding degradation efficiency of RhB (97.4%) and BPA (88.4%) after 140 min and 175 min of visible light irradiation, respectively, compared to pure BOC and other composite samples. The improved photocatalytic degradation performance could be ascribed to the synergetic effects of the larger active area of hierarchical microsphere, the extended light absorption to visible light range with Co_3_O_4_ NPs, and the suppression of e^−^-h^+^ recombination caused by the *p-n* junction formation of Co_3_O_4_/BOC.

## 2. Results and Discussion

### 2.1. Structural, Morphological, and Elemental Analyses

Figure 1 shows the morphology of the BOC fabricated with different volume ratios of ethylene glycol (EG) and ethyl alcohol (EtOH). BOC-1 (Figure 1a) synthesized in EG only and BOC-3 (Figure 1c) prepared in mixtures of EG and EtOH both have uniform 3D architectures, whereas BOC-2 (Figure 1b) prepared in pure EtOH solution shows a nanoparticle-like morphology with almost no agglomeration. Compared to BOC-1, the morphology of BOC-3 was more regular with a flower-like shape, and the size of the micro-flower was about 4 μm (Figure 1c). In addition, BOC-3 was also prepared with different solvothermal reaction times to understand the microspherical morphology growth and optimize the reaction time (discussed in Appendix A). BOC-3 synthesized following 6 h of solvothermal treatment was composed of many ultrathin nanosheets (inset of Figure 1), which is beneficial for photocatalytic degradation due to the increased specific surface area. Figure 1d shows the N_2_ adsorption-desorption isotherms of BOC-1 (dark), BOC-2 (red), and BOC-3 (blue). The isotherm plots showed type-IV isotherm and hysteresis loop curves [32]. The S_BET_ values of BOC-1, BOC-2, and BOC-3 were 8.914, 8.160, and 15.720 m^2^/g, respectively, indicating that BOC-3 had the largest specific surface area.

Figure 2a shows a FESEM image of the 20-Co_3_O_4_/BOC synthesized with 20 mg of Co_3_O_4_ NPs and BOC-3. Its hierarchical morphology was almost identical to that of BOC-3, and it was not affected by the incorporation of Co_3_O_4_ NPs during the synthesis process. To investigate the existence of the nanosized Co_3_O_4_ in the BOC hierarchical morphology, HRTEM measurements of 20-Co_3_O_4_/BOC were performed, as shown in Figure 2b. The results confirmed that the Co_3_O_4_ NPs, which had an approximate diameter of 10 nm, were decorated on the surface of BOC. In the HRTEM, the fringe spacing of 0.23 nm belonged to the (222) crystal plane of Co_3_O_4_. By contrast, the fringe spacings of 0.272 nm and 0.31 nm, which respectively refer to the (200) and (117) crystal planes of BOC, have also been observed. Further, a high-angle annular dark-field (HAADF) image of 20-Co_3_O_4_/BOC was obtained (Figure 2c), in which the tiny black spots identified across the BOC surface indicated Co_3_O_4_ NPs; this finding was further confirmed by EDX analysis in Figure 2d,e. These results confirmed the 0D/3D morphology of the prepared photocatalyst might be conducive to the photocatalytic performance of the Co_3_O_4_/BOC heterostructure. Moreover, the obtained EDS mapping spectrum (Figure A2 in Appendix B) confirmed the presence of bismuth (Bi), oxygen (O), chlorine (Cl), and cobalt (Co) elements in the 20-Co_3_O_4_/BOC sample. The at.% and wt.% of the elements are also shown (inset table in Figure A2). Figure 2f presents the N_2_ adsorption–desorption isotherm plots for 20-Co_3_O_4_/BOC. The estimated S_BET_ using N_2_ isotherms was found to be 14.873 m^2^/g, which was close to the S_BET_ value of BOC-3. This result shows that the surface area of 20-Co_3_O_4_/BOC was slightly affected by the incorporation of Co_3_O_4_ NPs.

The XRD patterns of the pristine BOC-3, Co_3_O_4_, and 20-Co_3_O_4_/BOC are shown in Figure 2g. The observed peaks in the XRD patterns of BOC-3 and Co_3_O_4_ matched the BOC and Co_3_O_4_ crystallites (JCPDS cards #37-0702 and #42-1467), respectively. The 20-Co_3_O_4_/BOC samples showed all characteristic peaks of BOC-3. However, the characteristic peaks of Co_3_O_4_ were not observed in the prepared composite samples. The absence of Co_3_O_4_ characteristic peaks was attributed to the low amount of Co_3_O_4_ compared to BOC in 20-Co_3_O_4_/BOC composite samples. The structural properties of 20-Co_3_O_4_/BOC were further investigated using Raman spectroscopy, as shown in Figure 2h. All characteristic peaks of BOC have been observed in the Raman spectrum of pure BOC-3 [33]. The observed peak at 165.80 cm^−1^ belonged to the A_1g_ internal stretching of the Bi-Cl bond [33,34]. Because of the oxygen-rich nature of BOC-3, the observed peaks in the range from 200 cm^−1^ to 500 cm^−1^ belong to the vibrational modes of Bi and O bonding. Among them, The peak at 470.51 cm^−1^ is the characteristic vibrational mode of BOC-3, which belongs to O-Bi-O bending modes. Further, the peak at 598.80 cm^−1^ represents Cl-Cl stretching modes [35]. Regarding Co_3_O_4_ NPs, all the characteristic peaks of Co_3_O_4_ appeared in the Raman spectra, indicating the successful formation of Co_3_O_4_ NPs, along with an extra peak at 481.15 cm^−1^ from the glass substrate. These observed peaks belonged to the F_2g_ and E_g_ modes of the combined vibrations of the tetrahedral site and octahedral oxygen vibrations [36,37]. The Raman spectra of Co_3_O_4_ NPs-decorated BOC were also obtained. The Raman spectra of 20-Co_3_O_4_/BOC showed all the characteristic peaks of both Co_3_O_4_ and BOC-3 samples, along with an extra peak at 307.50 cm^−1^. Since both Co_3_O_4_ and BOC-3 are oxygen-rich compounds, the interconnection of Co_3_O_4_ and BOC through oxygen bonding led to a new peak formation in the 20-Co_3_O_4_/BOC sample. The observed intense peak belongs to the Bi-O(1) rocking and weak O(2) breathing modes in the 20-Co_3_O_4_/BOC heterostructures, thus confirming the successful heterojunction formation [38].

To analyze the chemical composition and chemical state of the elements, we performed X-ray photoelectron spectroscopy (XPS) of the 20-Co_3_O_4_/BOC heterostructure photocatalyst (Appendix C). The XPS survey spectrum clearly demonstrated that all peaks were attributable to Bi, O, Cl, and Co elements, revealing that the heterostructure consisted of Bi, O, Cl, and Co elements, as shown in Figure A3 (See Appendix C). The high-resolution XPS spectra of Bi 4f, C 1s, O 1s, and Co 2p for the heterostructure are respectively shown in Figure A3b–e. The two strong peaks at 159.63 and 164.93 eV were assigned to Bi 4f_7/2_ and Bi 4f_5/2_, respectively, which are the features of Bi^3+^ in BOC (Figure A3b). As depicted in Figure A3c, the O 1s profile could be deconvoluted into three peaks, thus indicating the existence of three different kinds of O species in the sample. The peaks observed at 529.674 and 530.568 eV were assigned to the lattice oxygen metal bonds and hydroxyl (^•^OH) functional groups in 20-Co_3_O_4_/BOC, respectively [39]; the peak at 531.592 eV corresponded to oxygen vacancies in Co_3_O_4_ in the 20-Co_3_O_4_/BOC composite sample [40]. Figure A3d shows the spectrum of Cl 2p, which contained diverse peaks at 198.58 and 200.139 eV, respectively. These can be attributed to Cl 2p_3/2_ and Cl 2p_1/2_ of the Cl^−^ ions in the corresponding 20-Co_3_O_4_/BOC sample [41]. In Figure A3e, the Co 2p peak of 20-Co_3_O_4_/BOC showed Co 2p_3/2_ and Co 2p_1/2_ spin-orbit doublets. The peaks at 782.25 and 793.50 eV in 20-Co_3_O_4_/BOC corresponded to Co^2+^ ions, whereas the peaks observed at 779.50 and 792.50 eV were assigned to Co^3+^ ions, therefore indicating the coexistence of Co^2+^ and Co^3+^ in both samples [42].

### 2.2. Photocatalytic Performance

The photocatalytic performance of BOC-1, BOC-3, 10-Co_3_O_4_/BOC, 20-Co_3_O_4_/BOC, and 40-Co_3_O_4_/BOC was explored by degrading RhB dye in aqueous solution under visible light, as shown in Figure 3a,b. Figure 3a indicates that 20-Co_3_O_4_/BOC outperformed BOC-3, 10-Co_3_O_4_/BOC, and 40-Co_3_O_4_/BOC by decomposing RhB dye solution in 140 min. Further, the degradation rate of RhB in the presence of each photocatalyst could be determined by the pseudo-first-order kinetic model, as expressed in Equation (1).
ln C/C_0_ = kt,(1)
where k, C_0_, and C represent the reaction rate constant, initial concentration, and remaining concentration at time t, respectively. Figure 3b shows the reaction rate and degradation efficiency of all photocatalysts used for RhB degradation after 140 min. 20-Co_3_O_4_/BOC had the highest degradation rate of 2.21 × 10^−2^/min and an efficiency of 97.4%. Moreover, we compared the RhB photocatalytic degradation performance of the single BOC and 20-Co_3_O_4_/BOC photocatalyst with previously reported, similarly structured semiconducting photocatalysts, as shown in Table A1 (In Appendix E). Both synthesized BOC and Co_3_O_4_/BOC hierarchical microspheres possessed relatively higher degradation performance under similar test conditions. This enhanced efficiency might be related to the high surface area of hierarchically structured BOC and the formation of a Co_3_O_4_/BOC heterojunction with Co_3_O_4_ NPs-decoration.

Moreover, the photocatalytic degradation of BPA was performed to evaluate the photocatalytic activity of BOC and Co_3_O_4_/BOC, as shown in Figure 3c,d. The BPA degradation results showed that 20-Co_3_O_4_/BOC efficiently decomposed BPA aqueous pollutant solution in 170 min. Moreover, the degradation of BPA in the presence of each photocatalyst followed the 1st-order reaction kinetic model. Figure 3d shows the reaction rate and degradation efficiency after 110 min for all the photocatalysts used for BPA degradation. 20-Co_3_O_4_/BOC had the highest degradation rate of 1.66 × 10^−2^/min and an efficiency of 88.4%, followed by BOC-3, BOC-1, and BOC-2.

We further conducted a reusability test for the 20-Co_3_O_4_/BOC in the presence of RhB and BPA pollutants, as shown in Figure A4 (in Appendix D). During RhB and BPA degradation, a consistent decrease in the degradation of RhB and BPA occurred after the 3rd cycle of degradation, and its degradation performance was slightly reduced. This slight reduction in photocatalytic degradation may be attributable to the adsorption of RhB and BPA molecules on the surface of the 20-Co_3_O_4_/BOC sample.

### 2.3. Analysis of Enhanced Photocatalytic Activity of Co_3_O_4_/BOC

Photocatalytic activity is mainly attributed to light absorption capacity and the separation and transfer efficiency of photoinduced charge carriers. Firstly, photoluminescence (PL) measurements were conducted to investigate the recombination rate of photo-induced e^−^-h^+^ pairs in BOC-3 and 20-Co_3_O_4_/BOC photocatalysts, as shown in Figure 4a. The PL emission intensity of 20-Co_3_O_4_/BOC was lower than that of BOC-3, thus indicating reduced recombination of e^−^-h^+^ pairs in the 20-Co_3_O_4_/BOC photocatalyst. The transient photocurrent response was also measured to provide further support to the efficient separation of photo-generated charges, as shown in Figure 4b. The 20-Co_3_O_4_/BOC had a higher photocurrent response than BOC-3. The higher photocurrent response of 20-Co_3_O_4_/BOC was attributed to the higher separation efficiency of the excitons and its longer lifetime. Moreover, the EIS Nyquist plot was obtained to examine the electrode/electrolyte interfacial charge transfer resistance, as shown in Figure 4c. 20-Co_3_O_4_/BOC showed a smaller arc radius than BOC-3. The inset in Figure 4c shows the circuit of the sample-solution in the EIS measurements. According to the model of the circuit, R_s_ is related to uncompensated solution resistance, R_p_ is related to the porosity of the electrode, and R_ct_ represents the charge transfer resistance at the interface [43]. The R_ct_ values for BOC and 20-Co_3_O_4_/BOC are 1.58 × 10^−4^ Ω and 82.93 × 10^−4^ Ω, respectively. The smaller R_ct_ value of 20-Co_3_O_4_/BOC represents its lower charge transfer resistance, which is beneficial for high redox reactions during photocatalysis.

Further, the UV-vis DRS of Co_3_O_4_, BOC-3, and 20-Co_3_O_4_/BOC were measured to investigate the optical absorption ability, as shown in Figure 4d. Co_3_O_4_ showed absorption throughout the whole UV and visible range. The absorption edges for BOC-3 were located around 520 nm. In comparison, 20-Co_3_O_4_/BOC showed enhanced absorption in the visible range after loading Co_3_O_4_ NPs on BOC. The high absorption of 20-Co_3_O_4_/BOC was attributed to the strong contribution of Co_3_O_4_ in 20-Co_3_O_4_/BOC to the absorption of visible light. Ultimately, these results suggest that the formation of the heterojunction in 20-Co_3_O_4_/BOC heterostructure could effectively suppress the recombination of the photoexcited charge carriers and enhance the visible light absorption ability. Therefore, the photocatalytic performance could effectively be improved by the 20-Co_3_O_4_/BOC heterostructure.

### 2.4. Interfacial Charge Transfer Behavior and Photocatalytic Reaction Mechanism

The electronic structures of Co_3_O_4_, BOC, and Co_3_O_4_/BOC were analyzed by UV-Vis diffuse reflectance spectra (DRS), the Mott–Schottky (MS) plot, and valence band (VB) XPS measurements to elucidate the photocatalytic mechanism of the 20-Co_3_O_4_/BOC heterostructure during the photodegradation of RhB and BPA, as shown in Figure 5. First, the optical bandgap energies of BOC-3 and Co_3_O_4_ could be obtained through curve fitting of the Tauc plot of (αhν)^n/2^ versus hν (Figure 5a), where n = 4 for the Co_3_O_4_ direct band gap semiconductor and n = 1 for the indirect band gap semiconductor [44,45]. The obtained bandgap energies were 2.34 and 2.25 eV for BOC-3 and Co_3_O_4_, respectively.

Secondly, the Fermi energy (E_f_) levels of the prepared Co_3_O_4_ and BOC-3 were obtained using MS analysis, as shown in Figure 5b,c, respectively. The MS plots of Co_3_O_4_ and BOC-3 showed negative and positive slopes, thus indicating p-type and n-type semiconducting behaviors, respectively [46,47]. Further, by extrapolating MS plots, the flatband potentials of Co_3_O_4_ and BOC-3 were found to be +0.037 and −0.51 V, respectively, vs. the standard calomel electrode (SCE). Note that the flatband potential (E_fb_) of the n-type and p-type semiconductors represents the E_f_ level. The E_f_ level vs. the normal hydrogen electrode (NHE) scale could be calculated using Equation (2). The resultant E_f_ of Co_3_O_4_ and BOC-3 were +0.277 and −0.266 eV vs. NHE, respectively
E_fb_ (vs. NHE) = E_fb_ (vs. SCE) + 0.244 (eV),(2)

Valence band (VB) XPS measurement was conducted to determine the VB potentials of Co_3_O_4_ and BOC-3, as shown in Figure 5d. The VB XPS spectra revealed the VB maxima of 0.86 and 1.69 eV for Co_3_O_4_ and BOC-3, respectively. Thus, the VB potentials with respect to the E_f_ level were calculated to be 1.137 and 1.424 eV vs. NHE, respectively (Figure 6a). The CB minima of Co_3_O_4_ and BOC-3 were determined using the following Equation (3).
E_CB_ (vs. NHE) = E_VB_ (vs. NHE) − E_g_,(3)
where E_CB_, E_VB_, and E_g_ denote the sample’s CB potential, VB potential, and bandgap, respectively. As a result, the calculated CB potentials for Co_3_O_4_ and BOC-3 were −1.113 and −0.916 eV vs. NHE, respectively.

Therefore, based on the above analysis, we can obtain the band structures of Co_3_O_4_ and BOC in NHE scale before contact, as shown in Figure 6. When Co_3_O_4_ and BOC form the Co_3_O_4_/BOC heterojunction after they come into contact, the electrons will spontaneously migrate from BOC to Co_3_O_4_ through the Co_3_O_4_/BOC interface to align with the Fermi level because BOC has a higher E_f_ level than Co_3_O_4_. The migration of these electrons results in the band bending upward for BOC and downward for Co_3_O_4_, near the interface of BOC and Co_3_O_4_, respectively, as shown in Figure 6b. These band bendings lead to a depletion region at the Co_3_O_4_ and BOC-3 interface, thus resulting in the generation of an internal electric field (IEF) from BOC toward Co_3_O_4_ at the interface. After light is irradiated on the photocatalyst, the charge carriers simultaneously excite from VB to the CB in Co_3_O_4_ and BOC-3 to produce photo-generated e^−^ and h^+^ pairs. Then, the photoexcited electrons in the CB of Co_3_O_4_ will quickly migrate to the CB of BOC-3, whereas the remaining holes in the VB of BOC-3 will migrate to the VB of Co_3_O_4_ because of the IEF directed from BOC to Co_3_O_4_ in the 20-Co_3_O_4_/BOC, which is a typical charge transport of a type-II heterostructure. As a result, the e^−^-h^+^ recombination is suppressed in the heterostructure system. Moreover, the unique 0D/3D morphology of 20-Co_3_O_4_/BOC will provide more active catalytic reaction centers and increase the active sites. Thus, the developed heterostructure could be suggested to be beneficial in photocatalytic degradation.

Finally, photocatalytic active radical detection experiments were conducted to investigate the photocatalytic reaction mechanism and validate our proposed heterojunction formation, as depicted in Figure 7. Here, BQ, IPA, and KI were used as scavengers of ^•^O_2_^−^ radicals, ^•^OH radicals, and hole (h^+^), respectively, which are produced during the photocatalytic degradation of RhB [48]. Since 20-Co_3_O_4_/BOC composite decomposed the RhB dye efficiently compared to the other photocatalysts (shown in Figure 3a), a 20-Co_3_O_4_/BOC photocatalyst sample was chosen for active radical detection. As shown in Figure 7, in the presence of BQ and IPA, the degradation efficiency of RhB was significantly reduced from 98% (no scavenger) to 68.5% and 86.2%, respectively. However, no effect on the degradation efficiency was observed when using KI as an h^+^ scavenger. Therefore, ^•^O_2_^−^ and ^•^OH radicals were found to be the reactive species with increased generation of ^•^O_2_^−^ during the photodegradation of RhB.

Based on the radical scavenger experiments, the possible reaction mechanism was proposed as shown in reaction (4)–(10); the excited electrons on BOC reduced the oxygen molecule (O_2_) into ^•^O_2_^−^ radicals, and then the ^•^O_2_^−^ radicals reacted with e- and hydrogen ions (H^+^), ultimately resulting in the generation of hydrogen peroxide (H_2_O_2_) radical, which was further reduced to ^•^OH and hydroxyl ion (OH^−^). Then, the ^•^O_2_^−^, ^•^OH, and OH^−^ finally decomposed RhB and BPA into small chain molecules.
Co_3_O_4_ + hν → Co_3_O_4_* (e^−^ + h^+^)(4)
BOC + hν → BOC* (h^+^ + e^−^)(5)
Co_3_O_4_* (e^−^ + h^+^) + BOC^*^ (h^−^ + e^+^) → Co_3_O_4_^*^ (h^+^) + BOC* (e^−^)(6)
O_2_^−^ + e^−^ → ^•^O_2_^−^(7)
O_2_^−^ + e^−^ + 2H^+^→ H_2_O_2_^−^(8)
H_2_O_2_ + e^−^ + H^+^ → ^•^OH + OH^−^(9)
RhB/BPA + ^•^OH/OH^−^/O_2_^−^ → decomposed products(10)

## 3. Materials and Methods

### 3.1. Chemicals

Bismuth nitrate pentahydrate (Bi(NO_3_)_3_.5H_2_O, 99%), Cobalt acetate tetrahydrate (Co(CH_3_COO)_2_.4H_2_O), Potassium chloride (KCl, 99%), Ethylene glycol (EG) ((CH_2_OH)_2_, 99%), Dimethylformamide (DMF) (C_3_H_7_NO), Rhodamine-B (RhB) (C_22_H_24_N_2_O_8_, 99%), Bisphenol-A (BPA) (C_15_H_16_O_2_, 99%), 1,4-benzoquinone (BQ) (C_6_H_4_O_2_, 99%), and Fluorine doped tin oxide (FTO) glass were purchased from Sigma Aldrich Inc. (St. Louis, MO, USA). Ethanol (EtOH) (C_2_H_5_OH, 99%) and Isopropyl alcohol (IPA) (C_3_H_8_O, 99%) were purchased from DAEJUN Co., Ltd (Daejun, Korea). All reagents were used without any further purification.

### 3.2. Preparation of Co_3_O_4_ Nanoparticles

The Co_3_O_4_ NPs were prepared by the solvothermal method [29]. In a typical synthesis, 80 mg of Co(CH_3_COO)_2_.4H_2_O was dissolved in 60 mL of EtOH using a magnetic stirrer. The prepared solution was then transferred to a 100 mL Teflon-lined autoclave and heated at 150 °C for 4 h in a thermal oven to initiate the solvothermal reaction. After completion of the reaction, the autoclave was allowed to cool down to room temperature, at which point the raw Co_3_O_4_ was washed with EtOH and centrifuged at 15,000 rpm for 30 min. Following centrifugation, the collected sample was dried at 60 °C overnight to obtain the Co_3_O_4_ NPs.

### 3.3. Preparation of BOC and Co_3_O_4_/BOC

BOC was prepared by a solvothermal method, as shown in Figure 1a. In a typical synthesis method, 2.186 g (4.5 mmol) of Bi(NO_3_)_3_.5H_2_O was dissolved in 17.5 mL of ethanol by ultrasonication, followed by stirring. The resulting solution was referred to as solution-A. At the same time, 1.5 mmol of KCl was dissolved in 17.5 mL of EG by ultrasonication, followed by stirring for 30 min. The resulting solution was referred to as solution-B. Next, solution-B was added dropwise to solution-A and then constantly stirred for 30 min. Then, the resulting combined solution was transferred to a 50 mL Teflon autoclave and heated for 6 h at 160 °C. The product obtained in this way was washed and dried overnight at 60 °C. Afterward, the gray powder was collected and calcinated in a muffle furnace at 450 °C for 1 h to obtain the targeted BOC hierarchical microspheres. The Co_3_O_4_/BOC samples were prepared using the identical synthesis procedure with the addition of Co_3_O_4_ NPs (10, 20, and 40 mg) into the combined solution (see Figure 1b). Moreover, the BOC sample was prepared in pure EG and EtOH solutions to investigate the roles of EG, EtOH, and EG-EtOH mixed solvents during BOC synthesis. The samples prepared in EG, EtOH, and EG-ETOH mixed solvents were labeled as BOC-1, BOC-2, and BOC-3, and the sample prepared with the addition of 10, 20, and 40 mg of Co_3_O_4_ NPs were labeled as 10-Co_3_O_4_/BOC, 20-Co_3_O_4_/BOC, and 40-Co_3_O_4_/BOC, respectively.

### 3.4. Characterization of Samples

The crystal structure and phase purity were characterized by powder X-ray diffraction (XRD) in the 2θ range from 10~80° (2° min^−1^) using an X-ray diffractometer (Rigaku D/MAX-2500) with a Cu Kα irradiation source (λ = 1.54178 Å) and X-ray power of 40 kV/30 mA. Micro-Raman spectroscopy (XperRAM100, Nanobase Inc., Seoul, Korea) equipped with a monochromatic laser source (wavelength of 532 nm and power of 6 mW) was used to characterize the crystalline phase. The morphologies were examined using a field emission scanning electron microscope (SEM) (JSM-6700F, Jeol Ltd., Tokyo, Japan) and a transmission electron microscope (TEM) (“NEOARM “/JEM-ARM200F, Jeol Ltd.) equipped with an energy dispersive spectroscope (EDX). X-ray photoemission spectroscopy (XPS) (Veresprobe II, ULVAC-PHI Inc., Kanagawa, Japan) with a Monochromatic Al Kα X-ray source was used to examine the chemical compositions of the samples. The specific surface area was volumetrically assessed by measuring the nitrogen adsorption/desorption isotherms at 77 K using Microtrac, BELsorp-mini II. UV–vis diffuse reflectance spectra (DRS) were obtained using a spectrometer (V-750, Jasco Inc., Tokyo, Japan) equipped with a 60 mm integrating sphere while using BaSO_4_ as a reference. Photoluminescence spectra (PL) were collected using a spectrophotometer (FS5 fluorescence, Edinburg, United Kingdom) with an excitation wavelength of 375 nm. Electrochemical impedance spectroscopy (EIS) was performed using a three-electrode workstation (VSP Potentiostat, Biologic, Seyssinet-Pariset, France). A pt square plate (1 × 2 cm^2^) and a standard calomel were used as counter and reference electrodes, respectively. A clean fluorine-doped tin oxide (FTO) glass with an active surface area of 0.8 cm^2^ was used as a substrate for the working electrode, whereas an aqueous solution of 0.5 M Na_2_SO_4_ (80 mL) was used as the electrolyte. In the preparation of the working electrode, 0.1 mg of photocatalyst was added to 1 mL of DMF solution and sonicated for 1 h. Then, 50 μL of the dispersed solution was drop-casted on FTO and annealed at 160 °C for 1 h, which was further used in the electrochemical investigation.

### 3.5. Photocatalytic Activity Measurements

The photocatalytic characteristics of the samples were examined using a visible light source with a 300 W Xenon lamp (1000 W/m^2^) (CEL-HXF300, CEAULIGHT Co., Beijing, China) equipped with a UV-IR cutoff filter (420 nm > λ > 780 nm). A double wall jacket beaker with a surface area of 80 cm^2^ connected to a water chiller was used to perform the photocatalytic degradation measurement of the photocatalysts. The height from the surface of the pollutant solution to the light source was kept at 30 cm. In this study, RhB dye and BPA colorless pollutants were used to evaluate the degradation efficiency of the synthesized photocatalysts. Briefly, 20 and 40 mg of photocatalyst was used to degrade 40 mL of RhB dye (20 ppm) and BPA colorless pollutant (10 ppm), respectively. Before initiating the photocatalytic experiment, the RhB and BPA aqueous solutions were stirred for 60 min and 30 min, respectively, in dark conditions to attain an adsorption–desorption equilibrium of the photocatalysts. To investigate the photocatalytic degradation rate, 2 mL solution was taken from the RhB or BPA solution after a specific interval, and this solution was then centrifuged for 3 min at 5000 rpm to separate the photocatalyst, after which the absorbance spectrum of the supernatant using UV-visible spectrophotometer was measured. Further, radical trapping experiments were conducted to determine the dominant radical species involved in the photocatalytic decomposition of RhB and BPA. IPA, BQ, and KI of 2 mmol were used as trapping reagents to explore the active species, such as, ^•^O_2_^−^, ^•^OH radicals, and h^+,^ respectively.

## 4. Conclusions

In this work, we successfully developed a 3D hierarchical BOC microsphere and a 0D/3D-Co_3_O_4_/BOC heterojunction photocatalyst composed of BOC decorated with Co_3_O_4_ NPs using a simple solvothermal synthesis method. The developed heterostructure showed a conventional type II charge transport phenomenon across Co_3_O_4_ and BOC-3 by forming a *p-n* heterojunction. The 0D/3D hierarchical morphology of the Co_3_O_4_/BOC could increase active sites because of its high surface area, suppressing e^−^-h^+^ recombination, and improving visible light absorption. The results of mechanistic studies have proven that the generation of an IEF from BOC-3 to Co_3_O_4_ led to a *p-n* junction and the formation of a type II heterojunction. Thus, benefiting from the above properties, the Co_3_O_4_/BOC sample demonstrated a higher reaction rate and a higher degradation efficiency than bare Co_3_O_4_ and BOC during RhB and BPA degradation. Conclusively, our developed photocatalyst should be considered a good candidate for pollutant degradation.

## Data Availability

Not applicable.

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
