# Peer review of "Synthesis of Co3O4 Nanoparticles-Decorated Bi12O17Cl2 Hierarchical Microspheres for Enhanced Photocatalytic Degradation of RhB and BPA"

_ijms, 2022, doi:10.3390/ijms232315028_

Round 1

Reviewer 1 Report

Photocatlytic water cleaning is anticipated to capture a broad development space in the field of environmental sciences due to the use of freely available light. The high-performance photocatalytic materials are very critical. In this manuscript, the authors reported the “Synthesis of Bi12O17Cl2 hierarchical microspheres and Bi12O17Cl2/Co3O4 nanocomposite for photocatalytic degradation of RhB and BPA”. The author prepared a 3-dimensional hierarchical heterojunction, clearly explain the synthesis procedure, and investigate the role of each and heterojunction photocatalyst. The zero-dimensional Co3O4 nanoparticles decorated three dimensional Bi12O17Cl2 nanosheet like hierarchical morphology is seems interesting for the chemical reaction on the photocatalyst surface. The author provides strong evidence for the confirmation of Co3O4/Bi12O17Cl2 heterojunction formation based on mechanistic investigation, investigated the active radical during photocatalysis and its reaction mechanism. Therefore, I highly recommend this manuscript for publication in International Journal of Molecular Sciences after minor corrections. The following are my questions which are mostly related to English correction, which should be addressed before acceptance.

1.The English corrections required in the manuscript such as

A) (Bi2O2)2+ should be replaced with [Bi2O2]2+, (page 1, line#42)

B) The reference should be added after the author’s name in a sentence, for example, He et al. [ref], (page 2 line# 65 and line# 76)

C) In page 2, line# 70, lack of space between word and reference (i.e. p-n junction catalyst[18]).

D) 20-Co3O4/BOC synthesized with 20 g of Co3O4, is it correct? (Page 3, line#118)

2.The font size in some figures are too small. Need to increase font size for clear visibility. (For example, figure 2 (b, g, h), 3 (b, d) and figure 7.

3.In Figure 2 (h) a strong peak around 490 cm-1 is visible in the Raman spectra of Co3O4 and 20-Co3O4/BOC, which is not discussed. Please elaborate.

4.In Figure 4(c), the circuit diagram should be mentioned, and the resistance value should be calculated after fitting.

5.In figure 7, on the x-axis the “sample” is written to represent the scavengers, which is not suitable. The proper representation is required on the x-axis for better understanding.

Author Response

please, find attached response file.

Reviewer 2 Report

This work reported in this article could be interesting to the readers of the journal. However, I have several issues that should be fixed.

1-      Rewrite this sentence as is unclear: The enhanced performances are proposed to be 20 attributed to the synergetic effects of the high surface area of hierarchical BOC microsphere, 21 extended absorption of visible light with the Co3O4 NPs and suppression of electron-hole 22 recombination of Co3O4/BOC type-II heterostructure

2-      What is a “developed method” as written in this sentence: The developed method is beneficial ….

3-      I did not see any simulation results in this ms. All results are emerged from experiment. I am not sure whether the experimental data are reliable since they are not cross-checked with theory.

4-      The Raman modes are assigned, but not discussed. What is the significance of those modes? How do they help to understand the functionality of the systems examined?

5-      Several things are not consistent in this ms. For instance, the authors used terms “band gap” and “bandgap” randomly. Why?

6-      What are atomic and orbital origins of CB and VB? How did authors know whether the transition involved between them was direct or indirect?

7-      The ms needs English correction.

8-      The list of background references is not up to mark. There are many related references missing.

Author Response

please find attached response file

Reviewer 3 Report

The paper is not suitable for publication in IJMS because of the lack of novelty. Similar materials are known, and the reaction is too old. Degradation of RhB etc has been widely reported. The authors need to apply the material is some other reactions that are really significant for synthesis or environment protection.

Author Response

please find attached response files

Reviewer 4 Report

The authors have submitted an interesting article “Synthesis of Bi12O17Cl2 hierarchical microspheres and Bi12O17Cl2/Co3O4 nanocomposite for photocatalytic degradation of RhB and BPA" which thoroughly studies the synthesis and characterization of photoactive nanocomposites for organic dye degradation. The manuscript is well structured and reads well overall, although it will need some revisions. I suggest this article be published after a major revision.

*** General comments:

ü    The abstract is clear and concise and comprises all cornerstones including a brief/general introduction to the topic, a non-technical summary of the major findings, and their implications.

ü    The introduction is compelling, clear, and concise. The introduction part covers a proper description of the challenge/gap and a strong background in the field associated with a fair literature review, however, it can be improved further.

ü    The various sections of the body of the text are clear and concise overall.

ü    The experimental design is logical, however, there are still some comments to be covered and some concerns to be addressed.

ü    The conclusions are logical.

*** Suggested revisions:

1- First of all, I strongly recommend the authors provide a simple, high-quality, and informative “Graphical abstract” which can present the whole concept of your study at a glance. I would like to recommend authors design a “Graphical Abstract” for this study to better show the whole story in a simple and informative manner. In this regard, you can use illustrate a simple sketch of the big picture and add elements like Material synthesis pathway, material characterization (e.g., SEM images), and so forth (totally up to you).

2- Please carefully revise the manuscript to remove grammatical errors and vague sentences. Some of the sentences are unnecessarily long (like the very first sentence of the introduction) which makes it difficult and boring for the readers to follow them. Please double-check the whole manuscript and revise all.

* * I strongly suggest the author keep consistent in the figure's presenting style. The authors should select the same color and symbol for graph presentation which make readers easily follow data trend throughout the whole manuscript. Similarly, the y-axis of degradation graphs in figure 3 and many other similar errors.

3- The novelty statement of an article is of significant importance that highlights the importance of the current study and separates it from previously done research. In this work, the novelty statement poorly represents the work and the authors needed more development and better define their hypothesis and objectives and how the presented work differs from already, recently published reports in the field with similar concepts such as:

Facile synthesis of flower-like Bi12O17Cl2/β-Bi2O3 composites with enhanced visible light photocatalytic performance for the degradation of 4-tert-butylphenol- 10.1016/j.apcatb.2015.01.015

*** Moreover, among all aqueous pollutants (dyes, organic contaminants, heavy metals, pharmaceutical agents, etc.) why the authors have chosen these dyes for elimination? The reasoning for contamination selection should be briefly added to the introduction part.

4- One of the best ways to highlight the outcomes of a study is to tabulate a comparative master table to compare the findings of this study with recently published data from other research groups. I suggest the authors add a such table or update the existing one with data to compare the efficacy of different.

5- To increase the validity of obtained data, I suggest the researchers of this study repeat all experiments with at least 3 replications and do statistical analysis on the obtained data since it allows researchers to hold a degree of confidence that their findings are real, reliable, and not due to chance (for example figure 3 and so forth)

6- Some of the references in the introduction part are too old (e.g., 1999, etc.) and it is not acceptable at all. A myriad of research bodies has been published in recent years and you can find similar concepts and cite them in your paper rather than more than 3 decades old references. Moreover, in the introduction part to better present the fundamentals of the photodegradation process, please read and add valuable information from the following “key publication” as well:

https://doi.org/10.1016/B978-0-12-818806-4.00013-9 ,  https://doi.org/10.1016/B978-0-12-818806-4.00010-3

7- The reusability of photocatalysts is one of their most important criteria that should be studied. The authors should run a reusability test (up to 3 or 5 runs at least) and report the results (along with statistical analysis of course).

Author Response

please find attached response file

Round 2

Reviewer 2 Report

Authors have revised their paper based on my comments. However, they did not incorporate the simulation results in this study, which they want to publish in another paper. While their view goes over my head, I am not sure if the ms is worthy of publicaiton without simulation. It is better to add some priliminary results already gathered from simulations, which may support the experimental findings. 

Author Response

   As we response to your first review, our current manuscript focused on synthesis, characteristics and performance test, and possible photocatalytic mechanism of newly synthesized photocatalyst materials. However, we regretted that reviewer #2 is still only concerned about inclusion of the theoretical simulation results even though we provide point-to-point answers to reviewer’s comments in the first review stage. I do not think that all experimental papers should include theoretical calculation without any specific reasons if they provide experimental results enough to prove with various evidence. On the other hand, according to reviewer 2’s insistence, all theoretical paper should also include experimental results to verify.

   We agreed with your suggestion which we should include theoretical calculation to strengthen our manuscript scientifically. However, as you know, Bi12O17Cl2, especially Bi12O17Cl2/Co3O4 nanocomposite is a multi-atomic material system. Since the calculation processes take a lot of time-consuming, it cannot be completed in a few weeks. Secondly, our Lab mainly focuses on experimental research and does occasionally theoretically related calculation with theoretical group. Thus, we do not have the copyright of related simulation software. Therefore we cannot provide any simulation results within this short revision period. For theoretical simulation, we contacted with Prof. Weibin Zhang in Yunnan Normal University in China who is an expert in the simulation and calculation of photocatalyst. He agreed with collaboration of calculation of our materials’ system later.

  Please, first allow us to publish our experimental results in international journal of molecular sciences. Then, we will theoretically investigate the properties and potential application of our material system in details with theory group and promise to report further scientific knowledge later.  

Reviewer 3 Report

Although the authors tried to improve their manuscript be editing the languages, I still consider that it is not suitable for publication in the journal for the lack of novelty.

For the reasons please see the last round review report.

Author Response

  We provided a certificate of English editing from native speaker in English correction company. Please consider it.  

  As we mentioned our goal of this manuscript at the first review stage, we focused on the synthesis of a hierarchical microsphere structure of Bi and O-rich BiOCl among Bi-based oxyhalide materials as a new photocatalysis material with a high photocatalytic activity. Also, to further enhance the photocatalytic properties of the BiOCl, we designed and prepared heterostructure in which BiOCl was coupled with Co3O4 as a material for absorbing more light energy. Thus, design and synthesis of the microstructure and proper heterostructure and detailed characteristic study of the material system would be an enough novelty of our manuscript.

  Please allow us a chance to publish our manuscript in international journal of molecular sciences for reporting our results to photocatalysis research area.

Reviewer 4 Report

The manuscript is well-amended and all my concerns have been covered. I have no further comments.

Author Response

Thank you for your acceptance.